# The Acceleration and Deceleration Profiles of U-18 Women’s Basketball Players during Competitive Matches

**DOI:** 10.3390/sports7070165

**Published:** 2019-07-05

**Authors:** María Reina, Javier García-Rubio, José Pino-Ortega, Sergio J. Ibáñez

**Affiliations:** 1Research Unit “Performance Optimization and Sports Training”, Sport Science Faculty, University of Extremadura, 10004 Cáceres, Spain; 2Department of Physical Activity and Sport Science, Sport Science Faculty, University of Murcia, 30720 San Javier, Spain

**Keywords:** Acceleration, speed, Performance, inertial devices, quarter, playing position

## Abstract

The ability of a player to perform high-intensity actions can be linked to common requirements of team sports, and the ability to accelerate can be an important factor in successfully facing the opponent. The aim of this study was to determine the acceleration and deceleration profiles of U-18 women’s basketball players during competitive matches. This study categorized accelerations and decelerations by playing position and quarter. Forty-eight U-18 female basketball players from the same Spanish league participated in this study. Each player was equipped with a WimuPro^TM^ inertial device. Accelerations/decelerations were recorded. The number of accelerations and decelerations, intensity category, and type were recorded. These variables varied between quarters (first quarter, second quarter, third quarter, and fourth quarter) and playing positions (Guard, Forward and Center). The shorter but more intense accelerations took place in the last quarter, due to the tight results of the matches. Besides, players in the Guard positions performed more accelerations and their intensity was greater than that of other positions. An acceleration profile was established for the quarters of a basketball game, and was shown to depend on the playing position, being different for Guards, Forwards and Centers in U-18 women’s basketball players.

## 1. Introduction

Basketball is an intermittent high-intensity sport characterized by high aerobic and anaerobic demands. A major amount of energy depends on the aerobic pathway (aerobic glycolysis) [1,2]. However, high-intensity actions such as changes of direction, accelerations, decelerations, jumps, sprints, contacts, and other specific skills depend on the anaerobic pathway (anaerobic glycolysis). These actions are crucial in the athlete’s performance in team sports [2,3]. The understanding and identification of the energy demand required in basketball in specific positions is important to design training exercises and optimal game simulations, and thus to be able to develop and improve fitness programs with the aim of optimizing performance [4].

Previous research has reported on different analysis load models for training and competition, mainly through internal load analysis (physiological variables) and Time Motion Analysis (physical variables and movements). However, global positioning systems (GNSS), especially in football, are increasingly used to evaluate the external load [5]. Nevertheless, the impossibility of establishing communications between the device and satellites in closed spaces makes it difficult to use in indoor sports, due to the increase in statistical as well as factual error. Faced with this problem, radio frequency systems, known as Local Positioning Systems (LPM), are used, and their validity for the registration of parameters such as distance covered, maximum speed or acceleration, has been studied [6,7]. Currently, radiofrequency systems and Ultra-Wide-Band (UWB) technology are being used to record sports information in indoor conditions, due to their advantages such as the smaller size of the devices and the greater accuracy of the measurements [8].

Physical stress in training and competition is divided between internal and external load. External load represents the variables manipulated to induce internal stress and include acceleration and deceleration movements [9]. External measures provide an objective assessment of the players’ work output. Accelerometers have been used extensively in the general population as a measure of physical activity level [10]. The monitoring of the external load measurements derived from triaxial accelerometers is currently considered a viable tool in team sports. This device, allowing to record data in three planes, resembles the specific movements performed in basketball as the combination of defensive and offensive movements forward, backward and lateral [11].

The aim of this study was to analyze the number and duration of accelerations and decelerations, determining the profiles of U-18 women’s basketball players during competitive matches by an inertial device. Therefore, it was hypothesized that the acceleration’s length and intensity will decrease as the game progresses.

## 2. Materials and Methods

### 2.1. Participants

Forty-eight U-18 women’s basketball players from the same Spanish league participated in this study. The participants belonged to four different teams that participated in the final tournament. Each team played three matches (n = 144). Players were categorized according to their playing position (Guards: n = 13, 168.62 ± 5.94 cm; Forwards: n = 22, 176.87 ± 6.04 cm; Centers: n = 13, 183.77 ± 4.71 cm). All players and trainers were informed about the research protocol, requisites, benefits, and risks, and their written consent was obtained before the start of the study to conform to the Code of Ethics of the World Medical Association (Declaration of Helsinki), which was approved by the Ethics Committee of the University of Extremadura (no. 67/2017).

### 2.2. Measurements

Quantitative variables: The number of accelerations/decelerations was recorded. The variables analyzed in each acceleration/deceleration were Duration (ms), Start speed (km/h) and Acceleration peak (m/s^2^).

Qualitative variables: Accelerations and Decelerations were classified according to intensity into three ranges: A1 (Low: 1–2.5 m/s^2^); A2 (High: 2.5–4 m/s^2^); A3 (Sprint: >4 m/s^2^) and D1 (Low: −1–−2.5 m/s^2^); D2 (High: −2.5–−4 m/s^2^); D3 (Sprint: >−4 m/s^2^).

### 2.3. Design and Procedures

Each player was equipped with a WIMUPro^TM^ inertial device that was turned on and placed into a specific custom-made vest fitted tightly onto the back of the upper torso, as is typically used in games. SVIVO^TM^ software automatically analyzed all the data gathered by the inertial device and sent it to the computer screen in real time. This UWB system attempts to alleviate the reference problem of the satellites, using positioning techniques based on the time in which the signal propagates from the transmitter to the receiver. The UWB system was adjusted to the reference field before the start of the investigation, by going around the perimeter of the field so that it would be recognized as reference system 8. This system is composed of six antennas placed in a hexagon around the playing field (Scheme 1). The ANT + transmitter emits a wireless signal for several seconds and inertial devices include ANT + receivers that register a mark in the software when they receive a signal. This proposal enables the automatic synchronization of time and positioning data in the software (SPRO^TM^). The WIMUPro^TM^ inertial device, UWB system and Software come from the same organization (RealTrack Systems, Almería, Spain).

For the analysis of the competition, the UWB system was calibrated 1 h before the start of the games, and the WIMUPro^TM^ inertial devices were synchronized to the UWB system through the ANT + technology. Each player was equipped with the inertial device 20 min before the start of the match. In this way, there was a period of familiarization during the warm-up. Once the match started, total and live times were calculated using the SVIVO^TM^ software; with total time referring to all of the time that a player was on court, including all stoppages in play, but excluding breaks between quarters and times out. Live time corresponds to the time when the game clock was running and the player was on the court and also short moments in which the player was active during out-of–bounds.

### 2.4. Statistical Analysis

The distribution of the data was checked with the Kolmogorov-Smirnov test [12], to select the subsequent statistical analysis. A descriptive analysis of the data was performed with means and standard deviation of all the collected variables in the study in competition by quarters and playing position. A χ² and Cramer V were calculated with their level of significance to identify the differences among the qualitative variables. For the rest of the variables, a one-way ANOVA was performed. Differences between groups (quarters and playing position) were identified with the Bonferroni post-hoc test. The effect size according to Cohen’s d, was used to identify the differences between groups, considering effect sizes of <0.20 as trivial, 0.20–0.49 as small, 0.50–0.80 as medium, and >0.80 as large [13]. Magnitude-Based Inferences (MBI) were also calculated to assess the true value of an effect statistic [14]. In cases where the confidence interval overlapped with the threshold for substantial positive and negative values (±0.20 standardized units), the effect was considered unclear. Otherwise, the effects were deemed clear. Thereby, results were labelled as probabilities, probable, very probable and almost certain values being identified as the most important effects in practice. These describe if the value of the statistical results *p*, is important or not, in practice. Statistical analyses were performed using SPSS v.21 software (IBM, Inc, Chicago, IL, USA). Statistical significance was set at *p* < 0.05. For MBI calculations spreadsheets designed with this purpose were used [14].

## 3. Results

### 3.1. Results by Quarter

The number of accelerations and decelerations, and intensity category and type varied between quarters. The number of accelerations and decelerations declined from the first and third quarters to the second and fourth quarters (ACC: Q1 = 156.25; Q2 = 163.42; Q3 = 158.33; Q4 = 160.67) (DEC: Q1 = 153.00; Q2 = 154.75; Q3 = 148.00; Q4 = 156.67) but they were not statistically significant (χ² = 1.422; V Cramer = 0.010; *p* = 0.700). The second quarter is where the greatest number of accelerations was performed, and the last quarter is where the greatest number of decelerations was performed.

In terms of intensity ranges, a higher percentage of accelerations and decelerations was found in the lower range (A1 and D1). There were statistically significant differences in the intensity of the accelerations and decelerations between quarters (ACC: χ² = 39.608; V Cramer = 0.051; *p* = 0.000; DEC: χ² = 24.696; V Cramer = 0.073; *p* = 0.000), being higher in the last quarter compared to the rest (Figure 1).

The accelerations were longer in the first period (Q1 = 2138.31 ms; Q2 = 2157 ms) than the second period (Q3 = 2035.27 ms; Q4 = 2004.58 ms) with *Almost certain* magnitude-based inferences (*p* = 0.001; *MBI* = 99.8). The decelerations were longer in the first period (Q1 = 2005.64 ms; Q2 = 2033.90 ms) than the second period too (Q3 = 1897.33 ms; Q4 = 1912.57 ms) with *Almost certain* magnitude-based inferences (*p* = 0.001; *MBI* = 99.3–99.8). The accelerations start speed was greater in the first period (Q1 = 2.54 km/h; Q2 = 2.51 km/h) than the second period (Q3 = 2.34 km/h; Q4 = 2.41 km/h) but significant differences were found in the comparison with the third quarter (Q1: *p* = 0.003 and *MBI* = 98.1; Q2: *p* = 0.001 and *MBI* = 99.3). The deceleration’s start speed was greater in the first period (Q1 = 12.13 km/h; Q2 = 11.93 km/h) than the second period too (Q3 = 11.41 km/h; Q4 = 11.47 km/h) with *Almost certain* magnitude-based inferences (*p* = 0.001; *MBI* = 99.7–99.9). The minimum acceleration peak was achieved in Q3 (2.34 m/s^2^) compared to Q1 and Q2 (*p* = 0.003 and *MBI* = 99.3; *p* = 0.002 and *MBI* = 99.5) (Figure 2).

Table 1 shows in more detail the results of the analyses of the differences existing in the accelerations and decelerations according to the match period. The results are presented as possibilities, with a qualitative label. The labels with a value of probable, very probable and almost certain are those that identify important effects in practice.

MBI’s were used to evaluate the effect statistic [14]; the results labeled as “*Almost certain*” values were identified as the most important effects. “*Almost certain*” effects were observed in the length of accelerations and decelerations, being substantially higher in the first period (Q1 and Q2) compared to the second period (Q3 and Q4). In the start speed, there were important effects in decelerations, being higher in the first period compared to the rest of the periods.

### 3.2. Results by Playing Positions

The number of accelerations and decelerations, and intensity category and type varied between playing positions (χ² = 15.120; V Cramer = 0.032; *p* = 0.001). The Center position performed less accelerations and decelerations per minute than Guards and Forwards (ACC = Guard = 3.85 acc/min; Forward = 3.71 acc/min; Center = 2.97 acc/min) (DEC = Guard = 3.76 dec/min; Forward = 3.48 dec/min; Center = 2.59 dec/min). In terms of intensity ranges, the Guard position showed a greater time percentage in ranges A3 and D3, while Centers were the players who spent more time in the ranges of less intensity A1 and D1 (Figure 3). Therefore, there were statistically significant differences in the accelerations and decelerations intensity ranges between playing positions (ACC: χ² = 81.230; V Cramer = 0.073; *p* = 0.000; DEC: χ² = 95.632; V Cramer = 0.081; *p* = 0.000), being higher in Guards compared to the rest.

The results show statistically significant differences in the lengths of accelerations, maximum speed peak, and start speed (*p* < 0.005) among playing positions. There were differences in the lengths of accelerations with a low effect between Center and perimeter players (ACC = 2200.87 ms; DEC = 2050.87 ms), being greater in Guards and Forwards. The maximum speed peak and start speed were higher in Guards (ACC = 4.55 m/s^2^ and 2.70 km/h) than the rest of the playing positions with *almost certain* magnitude-based inferences (*p* = 0.001 and *MBI* = 99.8–99.9). In the variables of deceleration, no important effects were observed (Figure 4).

Table 2 shows in detail the results of the analyses of the differences among accelerations and decelerations performed by players according to their playing position.

The most important effects were found in guards, accelerations being more intense with higher start speed and maximum acceleration peaks compared to the rest.

## 4. Discussion

The objective of this study was to determine the accelerations and decelerations of U-18 women’s basketball players during competitive matches. This study categorized accelerations and decelerations by playing position, playing time and quarter. Significant differences were found according to quarter, playing positions and playing time. A greater number of acceleration and decelerations were performed in the second and last quarter, respectively. In the first period, accelerations were longer and the start speed was higher. Centers performed less accelerations and at a lower intensity than guards and forward. Currently, in the scientific field, great importance is being given to the investigation of variables that describe the acceleration and deceleration dimensions in intermittent sports activities [15]. Monitoring external load measurements derived from triaxial accelerometers is currently considered a viable tool in team sports. Authors have used triaxial accelerometer technology to determine the external load in basketball players during training and sports competition [11,16].

For a higher specificity in the analysis of the competition in basketball, the game is described based on period and playing position. In the analysis of game periods, no significant differences were found in the number of accelerations and decelerations performed by the players but their intensity changed, being higher in the last quarter. The shortest accelerations were observed in the last quarter (2004, 58 ms) probably due to fatigue; however, the acceleration peak and high start speed values were found in the last quarter. In women’s basketball competition, Delextrat et al. [5] found a decline in running and sprinting time in the last quarter compared to the rest of the periods; however, significant differences were not found [2,17,18]. In contrast, in male basketball, high-intensity actions decreased significantly in the fourth quarter, so it is important to differentiate between men and women. It is necessary to take into consideration that the results obtained are probably influenced by other situational variables (strategic decisions, the pace of the game, the structure of the game, etc.) especially at the end of a quarter or the game [1,19]. In this study, the top four teams in the league qualified for the final round. When best teams play against them, usually final score differences are balanced [20]. In addition, there are more interruptions in final quarters, due to free throws and time-outs [21]. Therefore, players have intermittent recovery during the pauses, performing higher intensity and shorter accelerations. Players perform longer accelerations at a higher speed in the first quarter. At the beginning of the game, teams try impose their playing style [21], attempting to increase the advantage as soon as possible [22].

Differences according to playing position are mainly due to the specialization of players and the demands of the competition [4,23]. In many studies, the difference has been noted between inside players and outside players [24]. Besides, players in these positions are clearly differentiated by their anthropometric characteristics in high-performance basketball [3], but not as clearly in trainees or non-professional players [25]. In this research, according to playing position, it was found that Centers performed the lower number of accelerations per minute and their intensity was lower in relation to the rest of the team. In fact, Guards need to perform at higher intensities longitudinally on the court; Forwards need to do the same but horizontally in their positional play; and the activity of the centers is restricted near the basket [4,5,26]. These variations in demands are evident among playing positions and physical performance [1,27]. Schelling and Torres [28] showed higher acceleration loads in the Guards. Smaller players have lower body mass, and therefore, accelerate applying a lower force. According to that, centers will have problems to accelerate because of their increased body mass, the acceleration being slower and at lower intensity and needing more time to achieve adequate speed. Guards and Forwards play more in the perimeter. They have to perform different changes of directions, surprising movements, and changes of speed as backdoors, in order to destabilize their opponents. These playing patterns result in more and higher intensity accelerations and decelerations in these players compared with centers.

Monitoring competition and reporting data individually appears to be essential for designing specific training sessions for the competitive demands of each player [29]. The identification of individual acceleration profiles would help coaches and sports scientists to develop specific position-dependent exercises to improve players’ conditioning. Some players may reach a higher volume of work for the entire session, while others may do less work overall but consistently reach higher intensities.

## 5. Conclusions

An acceleration profile has been established for the first, second, third, and fourth quarter of a basketball game, as well as playing position, obtaining an acceleration profile in Guards, Forwards and Centers in U-18 women’s basketball players. An understanding of these differences would make it possible to design training sessions that are adequate for competitive demands. Therefore, specificity and individualized training principles have been shown to be important to assess performance and acceleration profile. As a future prospect, it would be vital to increase the sample, collecting a greater number of matches from different categories to observe the differences in the acceleration profiles in formative stages.

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
