# Peer review of "The Acceleration and Deceleration Profiles of U-18 Women’s Basketball Players during Competitive Matches"

_sports, 2019, doi:10.3390/sports7070165_

Reviewer 1 Report

General Comments

Thank you for the opportunity to review this manuscript, which has the potential to add to the Basketball research. There are some areas which require attention:

·         The Introduction needs to be a little more comprehensive and include some additional literature regarding the movement characteristics and demands of basketball.

·         The results need to be re-written so that the descriptive and inferential sections are combined

·         The Discussion section requires additional detail and comparison to the results of previous studies

Specific Comments

Line 16: ‘collaboration’ appears to be the wrong word to use here

Line 23: How did they vary, can you be more specific, was there an increase or decrease?

Lines 34-35: ‘…like other collective sports…’ needs to be rephrased for clarity.

Line 58: ‘This device allow measure in…’ is not correct English, please amend accordingly.

Line 59: Please aid the aims and hypotheses for this investigation

Line 64: Please add the physical characteristics of these positional groups

Lines 71-72: The unit of measurement should be either m/s/s or m.s2 please amend throughout

Line 103: Delete ‘Next’

Line 103: What post-hoc analysis was used along with the ANOVA?

Line 104: How were the Cohen’s d values interpreted, please provide an appropriate scale?

Results:

Sections 3.1 and 3.3 should be merged with sections 3.2 and 3.4 also merged. Currently this is too long and disjointed. When making these revisions please also make it clear if the differences were an increase or decrease and if they were significant and / or meaningful, based on the results of your statistical analyses.

Discussion:

The initial paragraph is too vague and should summarise the key findings.

There are also a series of very short paragraphs which should be either expanded, or some of them merged, ass appropriate. Greater comparison of your results to those in other studies is also required.

Line 194: Were these changes increases or decreases, or were there no clear trends?

Lines 200-202: Please re-word as appropriate as this does not appear to be substantiated by your results. This is currently too speculative.

Line 203: Change ‘in’ to ‘into’

Line 217: This should be (p<0.001)< p=""

Author Response

Open Review 1

English language and style

( ) Extensive editing of English language and style required 
(x) Moderate English changes required 
( ) English language and style are fine/minor spell check required 
( ) I don't feel qualified to judge about the English language and style 

Yes

Can be improved

Must be improved

Not applicable

Does the introduction provide sufficient   background and include all relevant references?

( )

( )

(x)

( )

Is the research design appropriate?

( )

( )

(x)

( )

Are the methods adequately described?

( )

( )

(x)

( )

Are the results clearly presented?

( )

( )

(x)

( )

Are the conclusions supported by the   results?

( )

( )

(x)

( )

Comments and Suggestions for Authors

General Comments

Thank you for the opportunity to review this manuscript, which has the potential to add to the Basketball research. There are some areas which require attention:

·         The Introduction needs to be a little more comprehensive and include some additional literature regarding the movement characteristics and demands of basketball.

·         The results need to be re-written so that the descriptive and inferential sections are combined

·         The Discussion section requires additional detail and comparison to the results of previous studies

 Specific Comments

Line 16: ‘collaboration’ appears to be the wrong word to use here

Thanks for the comment. We have been corrected it in the text. we have replaced the word with collaborative

Line 23: How did they vary, can you be more specific, was there an increase or decrease?

In general, it is important to know that the variables analyzed vary according to the period and the position. In the section on results, it is more specifically detailed how these variations are. For example: “The number of accelerations and decelerations declined from the first and third quarters to the second and fourth quarters”; “The second quarter is where the greatest number of accelerations was performed”; “being higher in the last quarter” or “The accelerations were longer and the start speed was greater in the first period”

Lines 34-35: ‘…like other collective sports…’ needs to be rephrased for clarity.

The sentence has been modified: “However, explosive actions such as changes of direction, accelerations, decelerations, jumps, sprints, contacts, and specific skills depend of the anaerobic pathway. They are determinants mainly in collective sports and in the final performance of the athletes”

Line 58: ‘This device allow measure in…’ is not correct English, please amend accordingly.

The sentence has been modified: This device measure in three planes, what resembles the specific movements performed in basketball as the combination of defensive and offensive movements forward, backward and lateral”

Line 59: Please aid the aims and hypotheses for this investigation

The aim and hypotheses have been aid: “For all this, to the best knowledge, the aim of this study was to determine the acceleration and deceleration profiles of U-18 women’s basketball players during competitive matches, using an inertial device” “It was hypothesized that the greatest number of accelerations and decelerations will be performed in the less intense ranges. Second, the duration and intensity of the accelerations will decrease as the game progresses”

Line 64: Please add the physical characteristics of these positional groups

The average height and standard deviation for each specific position has been added: with specific characteristics (Guard: 168,62±5,94 cm; 176,87±6,04 cm; Center: 183,77±4,71 cm)

Lines 71-72: The unit of measurement should be either m/s/s or m.s2 please amend throughout

Thanks for the comment. We have been corrected it in the text

Line 103: Delete ‘Next’

Thanks for the comment. We have deleted it in the text

Line 103: What post-hoc analysis was used along with the ANOVA?

We hace used Bonferroni post-hoc. We have added it in the text.

Line 104: How were the Cohen’s d values interpreted, please provide an appropriate scale?

Thanks for the comment. We have added this sentence in the text: “The effect size according to Cohen’s d, was used to identify the differences between groups where effect sizes of 0.20 are small, 0.50 are medium and 0.80 are considered large” (Thalheimer and Cook, 2002)

Results:

Sections 3.1 and 3.3 should be merged with sections 3.2 and 3.4 also merged. Currently this is too long and disjointed. When making these revisions please also make it clear if the differences were an increase or decrease and if they were significant and / or meaningful, based on the results of your statistical analyses.

Thanks for the comment. We thought it was a great contribution to improve our work. We have tried to combine the descriptive and inferential results according to the period and the playing position

Discussion:

The initial paragraph is too vague and should summarise the key findings. There are also a series of very short paragraphs which should be either expanded, or some of them merged, ass appropriate. Greater comparison of your results to those in other studies is also required.

We have to answer both reviewers’ suggestions at the same time. Both reviewers suggest similar ideas and corrections.

As reviewer 1 stated “The initial paragraph is too vague and should summarize the key findings”.  We have modified this paragraph, being more specific in the key findings. And as reviewer 2 stated “Please modify as to provide possible explanations.” We also have modified and provide more explanations in following paragraphs:

“For a greater specificity in the analysis of the competition in basketball, the investigations describe the game based on the period and the playing position. In the case of the match periods, this research found no significant differences in the number of accelerations and decelerations performed between quarters but their intensity changed. The shortest accelerations were produced in the last quarter (2004, 58 ms), this could be due to fatigue but the maximum acceleration velocity and start velocity values were also found here. Delextrat et al. [5] analyzing the time-motion analysis in competition where players decrease the time running and sprinting in the last quarter, but no statistically significant differences were found in the performance of high intensity actions [2,15,16]. On the contrary, with data from male players the high intensity movements decreased in the 4th quarter. This advocates that training high intensity skills might be important in male’s basketball but not in women [5], finding the need to establish differences between genders in training. It is necessary to take into consideration that the results obtained are probably influenced by other contextual factors (strategic decisions, the pace of the game, the structure of the game, etc.) especially in the end of a quarter or of the game [1,17]. In this study, the four best teams in the league classify to a final round. When bets teams plays against them, usually final score differences are balanced (Garcia et al, 2013). In addition, is in the final quarters of balanced games where more interruptions occurs due to free throws and time-outs (García-Rubio et al 2015). These can lead players to have intermittent recovery during these pauses, allowing perform high intensity accelerations, but shorter ones. Players perform long accelerations at higher speed in the first quarter, due to at the beginning of the games, teams try to perform better than their opponents do, in order to impose their style of play (García-Rubio et al 2015), trying to increase the advantage aa soon as possible (Marcelino et al., 2012).”

AND

Differences according to playing position are mainly due to the specialization of players and the requirements of the competition [4,18]. In many studies the difference has been noted between inside players and outside players [19]. Besides, players in these positions are clearly differentiated by their anthropometric characteristics in high-performance basketball [3], but not as clearly in trainees or non-professional players [20]. In this research, according to the playing position, it was found that Centers performed a lower number of accelerations per minute and their intensity was lower in relation to the rest of the team. Guards need to perform at higher intensities from basket to basket; Forwards need to do the same and from side to side during positional play; Centers activity is constraint, mostly, to the paint (Sampaio et al, 2008; Reina et al, 2018; Delextrat et al., 2015). These variations in demands are evident among playing positions and levels of anaerobic and aerobic fitness [1,21]. Centers are usually the bigger players on the teams, with the guards being the smallers ones. Schelling and Torres [22] showed higher acceleration loads for guards. These authors justified this finding stating that the smaller the player, the smaller the body mass and the easier to accelerate with less force applied. According to that, centers will have problems to accelerate because of their increase body mass, being this acceleration slower and at lower intensity, during more time to achieve the adequate speed. Guards and Forwards play more in the perimeter, and they have to perform different changes of directions, surprising movements with and without the ball, changes of speed as backdoors, in order to destabilize their opponents. These patterns of play lead to perform more and at high intensity accelerations and decelerations than centers.

Line 194: Were these changes increases or decreases, or were there no clear trends?

We add this statement obtained in the results at the end of the sentece: “being higher in the last quarter

Lines 200-202: Please re-word as appropriate as this does not appear to be substantiated by your results. This is currently too speculative.

We have added some references abour this: 2, 5, 12, 18.

We tried to explain that in the literature there were no differences in high intensity actions throughout the game by women's basketball. However, they are found in men's basketball, since at the end of the match the high intensity actions are minor. Therefore, it is considered necessary to work on these differences.

Line 203: Change ‘in’ to ‘into’

Thanks for the comment. We have been corrected it in the text

Line 217: This should be (p<0.001)< span="">

Thanks for the comment. We have been corrected it in the text

Reviewer 2 Report

GENERAL

The aim of this study was to determine es. This 18 study categorized accelerations and decelerations by playing position and quarter. Forty-eight U-19 18 female basketball players from the same Spanish league participated in this study. Each player 20 was equipped with a WimuProTM inertial device. Accelerations / decelerations were recorded. The 21 number of accelerations and decelerations, intensity category and type were recorded. These 22 variables varied between quarters (First quarter, second quarter, third quarter, fourth quarter) and 23 playing positions (Guard, Forward and Center). The shorter but more intense accelerations took 24 place in the last quarter, due to the tight results at the ends of the matches. Besides, players in the 25 Guard positions performed more accelerations and their intensity was greater than that of other 26 positions. An acceleration profile was established for the quarters of a basketball game, and was 27 shown to depend on the playing position, being different for Guards, Forwards and Centers in U-28 18 women’s basketball players.

This study aimed to examine the acceleration and deceleration profiles of U-18 women’s basketball players during competitive matches, using an inertial device.

The number of accelerations and decelerations, as well as their magnitude and type, varied between quarters (First quarter, second quarter, third quarter, fourth quarter) and playing positions (Guard, Forward and Center). The shorter but more intense accelerations took place in the last quarter, due to the tight results at the ends of the matches. Guard positions performed more accelerations and their intensity was greater than that of other positions.

This is an interesting descriptive study, as it presents the acceleration/deceleration profiles of young female basketball players. However, the authors should justify what this study adds to already published data from their own group and from others (e.g. Pino-Ortega et al., Front Psychol . 2019 May 15;10:1100. doi: 10.3389/fpsyg.2019.01100. eCollection 201.

Language must be thoroughly checked and corrected throughout the manuscript. Also expressions such as “aerobic route”, “collective sports” should be checked and corrected

There is no clear purpose of the study stated at the end of the Introduction. Please add.

Please delete or rephrase the text in lines 158-160, explaining the statistical approach to the reader. Readers are aware of the differences between p values and inferential statistics. Do not guide the reader to what you support. For example in Table 1, the p value for the Q1-Q2 difference is 0.893, while the “Magnitude” is described as “Possible”. Please amend accordingly.

The Results in Tables 1 and 2 should be summarized in the Text.

The first (and largest) part of the Discussion repeats the findings. Please modify as to provide possible explanations.

Lines 214-214: This statement referring to anaerobic and aerobic fitness is too general and not substantiated. Please amend.

The conclusion supports that an acceleration profile for U18 female basketball players was made. However, the limitations of the study should be clearly stated here (e.g. low number of matches used, influence of opponents, etc)

Minor comments

Fig 1. Please explain what A1, A2 and A3 are in the Figure legend

Fig. 2. Please explain what exactly is “Speed during decelerations” Are the units correct? (m/s or Km/h)? Why are they different from accelerations?. Also, the two black lines representing Max and start speed are difficult to identify (which is which) in the Figure. The same should be done in Fig 4.

Author Response

Open Review 2

English language and style

( ) Extensive editing of English language and style required 
(x) Moderate English changes required 
( ) English language and style are fine/minor spell check required 
( ) I don't feel qualified to judge about the English language and style 

Yes

Can be improved

Must be improved

Not applicable

Does the introduction provide sufficient   background and include all relevant references?

( )

( )

(x)

( )

Is the research design appropriate?

( )

(x)

( )

( )

Are the methods adequately described?

( )

(x)

( )

( )

Are the results clearly presented?

( )

(x)

( )

( )

Are the conclusions supported by the   results?

( )

( )

(x)

( )

Comments and Suggestions for Authors

GENERAL

The aim of this study was to determine es. This 18 study categorized accelerations and decelerations by playing position and quarter. Forty-eight U-19 18 female basketball players from the same Spanish league participated in this study. Each player 20 was equipped with a WimuProTM inertial device. Accelerations / decelerations were recorded. The 21 number of accelerations and decelerations, intensity category and type were recorded. These 22 variables varied between quarters (First quarter, second quarter, third quarter, fourth quarter) and 23 playing positions (Guard, Forward and Center). The shorter but more intense accelerations took 24 place in the last quarter, due to the tight results at the ends of the matches. Besides, players in the 25 Guard positions performed more accelerations and their intensity was greater than that of other 26 positions. An acceleration profile was established for the quarters of a basketball game, and was 27 shown to depend on the playing position, being different for Guards, Forwards and Centers in U-28 18 women’s basketball players.

This study aimed to examine the acceleration and deceleration profiles of U-18 women’s basketball players during competitive matches, using an inertial device.

The number of accelerations and decelerations, as well as their magnitude and type, varied between quarters (First quarter, second quarter, third quarter, fourth quarter) and playing positions (Guard, Forward and Center). The shorter but more intense accelerations took place in the last quarter, due to the tight results at the ends of the matches. Guard positions performed more accelerations and their intensity was greater than that of other positions.

This is an interesting descriptive study, as it presents the acceleration/deceleration profiles of young female basketball players. However, the authors should justify what this study adds to already published data from their own group and from others (e.g. Pino-Ortega et al., Front Psychol . 2019 May 15;10:1100. doi: 10.3389/fpsyg.2019.01100. eCollection 201.

 Thanks for the comment. We try to justify the need of this study in relation with Pino et al. (2019) and the rest of the literature. First big difference is the sample. In our study, a sample of females u´18 basketball players of national level is analyzed. In Pino et al. (2019), participants are U´18 males of international level. Characteristics and demand of basketball players are differences as it is have been probed, in fitness (Ramos, S., Volossovitch, A., Ferreira, A. P., Barrigas, C., Fragoso, I., & Massuça, L. (2019). Differences in maturity, morphological, and fitness attributes between the better-and lower-ranked male and female u-14 Portuguese elite regional basketball teams. The Journal of Strength & Conditioning Research) or in performance (Sampaio, J., Godoy, S. I., & Feu, S. (2004). Discriminative power of basketball game-related statistics by level of competition and sex. Perceptual and motor Skills, 99(3_suppl), 1231-1238; Madarame, H. (2018). Age and sex differences in game-related statistics which discriminate winners from losers in elite basketball games. Motriz: Revista de Educação Física, 24(1)).

Also, and more important, Pino et al (2019) studying external load of competitive matches, performing a quantitative analysis of different game demands as accelerations and decelerations, distance or player Load. Our study aim to perform a qualitative analysis of accelerations and decelerations, its duration, intensity and start duration. As Mara et al stated (Mara, J. K., Thompson, K. G., Pumpa, K. L., & Morgan, S. (2017). The acceleration and deceleration profiles of elite female soccer players during competitive matches. Journal of science and medicine in sport, 20(9), 867-872.), findings from this study can be used to develop match-specific acceleration and deceleration drills to optimise change of speed ability.

Language must be thoroughly checked and corrected throughout the manuscript. Also expressions such as “aerobic route”, “collective sports” should be checked and corrected

Thanks for the comment. We have been corrected it in the text. We have replaced collective with collaborative and route with pathway.

There is no clear purpose of the study stated at the end of the Introduction. Please add.

The aim and hypotheses have been aid: “For all this, to the best knowledge, the aim of this study was to determine the acceleration and deceleration profiles of U-18 women’s basketball players during competitive matches, using an inertial device” “It was hypothesized that the greatest number of accelerations and decelerations will be performed in the less intense ranges. Second, the duration and intensity of the accelerations will decrease as the game progresses”

Please delete or rephrase the text in lines 158-160, explaining the statistical approach to the reader.

OK, Done.

Readers are aware of the differences between p values and inferential statistics. Do not guide the reader to what you support. For example in Table 1, the p value for the Q1-Q2 difference is 0.893, while the “Magnitude” is described as “Possible”. Please amend accordingly.

We do not guide reader to what we support. The magnitude is labeled as “Possible” according to the Magnitude Based Inferences (MBI), that is complementary to p-values. As Sullivan and Feinn (Sullivan, G. M., & Feinn, R. (2012). Using effect size—or why the P value is not enough. Journal of graduate medical education, 4(3), 279-282.) stated:

“The P-value is the probability, when the null hypothesis is true (eg, no difference or no association), of obtaining a result equal to or more extreme than what we actually observed. Simplistically, P-value quantifies the probability that the result is due to chance. It does not measure how big the association or the difference is. The CI on a value describes the probability that the true value is within a given range. A 95% CI means that the CI covers the true value in 95 of 100 performed studies. The test is significant if the CI does not include the null hypothesized difference or association (eg, 0 for difference). The effect-sizes are quantitative measures of the strength of a difference or association. If the P-value is<0.05 but the effect size is very low, the test is statistically significant but probably, clinically not so. Scientific publications require more parameters than a P-value. Statistical results should also include effect sizes and CIs to allow for a more complete, honest, and useful interpretation of scientific findings

 We have tried to be more accurate in our analysis because of the p-value controversy, adding effects sizes and MBI. However, of course, we have tried to make it clear in the text. The following text were included:

“Magnitude Based Inferences (MBI) was also calculated. MBI helps to make a decision about the true value of an effect statistic (Batterham & Hopkings, 2006). Inferences about magnitudes were mechanistic: if the confidence interval overlapped thresholds for substantial positive and negative values (±0.20 standardized units), the effect was deemed unclear; effects were otherwise deemed clear. Based on that, results are presented as possibilities, with a qualitative label. These describe if the value of the statistical results p, is important or not, in practice. The labels with a value of probable, very probable and almost certain are those that identify important effects in practice. Statistical analyses were performed using SPSS v.21 software (IBM, Inc, Chicago, IL, USA). Statistical significance was set at p < .05. For MBI spreadsheets designed with this purpose were used (Batterham & Hopkings, 2006).”

The Results in Tables 1 and 2 should be summarized in the Text.

Has been added:

Table 1: There are statistically significant differences between quarters but it is the magnitude analysis that evaluates whether the values are important or not, in practice. “Almost certain” values were identified like the most important effects. Principally, these effects were observed in the duration of accelerations and decelerations, being much higher in the first period (Q1 and Q2). Specifically, in decelerations there were important effects in the start speed being higher in the first period too”

Table 2: The most important effects were found between the players in the guard position and the rest of the positions. The accelerations of the guards during each match were more intense. Accelerations with a start speed and maximum peaks of acceleration performed by guard were superiors to the rest.”

The first (and largest) part of the Discussion repeats the findings. Please modify as to provide possible explanations.

 We have to answer both reviewers’ suggestions at the same time. Both reviewers suggest similar ideas and corrections.

As reviewer 1 stated “The initial paragraph is too vague and should summarize the key findings”.  We have modified this paragraph, being more specific in the key findings. And as reviewer 2 stated “Please modify as to provide possible explanations.” We also have modified and provide more explanations in following paragraphs:

“For a greater specificity in the analysis of the competition in basketball, the investigations describe the game based on the period and the playing position. In the case of the match periods, this research found no significant differences in the number of accelerations and decelerations performed between quarters but their intensity changed. The shortest accelerations were produced in the last quarter (2004, 58 ms), this could be due to fatigue but the maximum acceleration velocity and start velocity values were also found here. Delextrat et al. [5] analyzing the time-motion analysis in competition where players decrease the time running and sprinting in the last quarter, but no statistically significant differences were found in the performance of high intensity actions [2,15,16]. On the contrary, with data from male players the high intensity movements decreased in the 4th quarter. This advocates that training high intensity skills might be important in male’s basketball but not in women [5], finding the need to establish differences between genders in training. It is necessary to take into consideration that the results obtained are probably influenced by other contextual factors (strategic decisions, the pace of the game, the structure of the game, etc.) especially in the end of a quarter or of the game [1,17]. In this study, the four best teams in the league classify to a final round. When bets teams plays against them, usually final score differences are balanced (Garcia et al, 2013). In addition, is in the final quarters of balanced games where more interruptions occurs due to free throws and time-outs (García-Rubio et al 2015). These can lead players to have intermittent recovery during these pauses, allowing perform high intensity accelerations, but shorter ones. Players perform long accelerations at higher speed in the first quarter, due to at the beginning of the games, teams try to perform better than their opponents do, in order to impose their style of play (García-Rubio et al 2015), trying to increase the advantage aa soon as possible (Marcelino et al., 2012).”

AND

Differences according to playing position are mainly due to the specialization of players and the requirements of the competition [4,18]. In many studies the difference has been noted between inside players and outside players [19]. Besides, players in these positions are clearly differentiated by their anthropometric characteristics in high-performance basketball [3], but not as clearly in trainees or non-professional players [20]. In this research, according to the playing position, it was found that Centers performed a lower number of accelerations per minute and their intensity was lower in relation to the rest of the team. Guards need to perform at higher intensities from basket to basket; Forwards need to do the same and from side to side during positional play; Centers activity is constraint, mostly, to the paint (Sampaio et al, 2008; Reina et al, 2018; Delextrat et al., 2015). These variations in demands are evident among playing positions and levels of anaerobic and aerobic fitness [1,21]. Centers are usually the bigger players on the teams, with the guards being the smallers ones. Schelling and Torres [22] showed higher acceleration loads for guards. These authors justified this finding stating that the smaller the player, the smaller the body mass and the easier to accelerate with less force applied. According to that, centers will have problems to accelerate because of their increase body mass, being this acceleration slower and at lower intensity, during more time to achieve the adequate speed. Guards and Forwards play more in the perimeter, and they have to perform different changes of directions, surprising movements with and without the ball, changes of speed as backdoors, in order to destabilize their opponents. These patterns of play lead to perform more and at high intensity accelerations and decelerations than centers.

Lines 214-214: This statement referring to anaerobic and aerobic fitness is too general and not substantiated. Please amend.

 Ok, we have substituted these concepts for “physical performance” because with the interpretation of the results we can establish parameters of máximum perfromance in this categroy and sports.

The conclusion supports that an acceleration profile for U18 female basketball players was made. However, the limitations of the study should be clearly stated here (e.g. low number of matches used, influence of opponents, etc)

Has been added: Therefore, specificity and individualized training principles have been shown to be important to match performance and acceleration profiling. As a future prospect it would be vital to increase the sample and collect greater number of matches belonging to different categories. And, Consequently, to be able to observe the differences in the profile of acceleration according to the formative stage.”

 Minor comments

 Fig 1. Please explain what A1, A2 and A3 are in the Figure legend

Thanks for the comment, the legend has been added: (Low: 1-2.5 m/s2); A2 (High: 2.5-4 m/s2); A3 (Sprint: > 4 m/s2) and D1 (Low: -1- -2.5 m/s2); D2 (High: -2.5- -4 m/s2); D3 (Sprint: > - 4 m/s2)

Fig. 2. Please explain what exactly is “Speed during decelerations” Are the units correct? (m/s or Km/h)? Why are they different from accelerations?. Also, the two black lines representing Max and start speed are difficult to identify (which is which) in the Figure. The same should be done in Fig 4.

Thanks for the comment. We have divided the graphs for a better understanding of the variables and their units of measures

Round  2

Reviewer 1 Report

The authors have made substantial revisions to the manuscript; however, the use of English and the scientific style of writing require attention throughout. Unfortunately, the errors relating to the scientific style of writing and use of English make this difficult to follow and result in some key points being lost.  Please see specific comments below.

Line 31: The term ‘explosive’ should not be used, please see, Winter et al. (2016). Misuse of "Power" and other mechanical terms in Sport and Exercise Science Research. The Journal of Strength & Conditioning Research 30(1): 292-300.

Line 36: Change ‘of’ to ‘on’

Line 60: Delete, ‘For all this, to the best knowledge…’ as it is redundant and makes no sense as part of this sentence.

Line 63-63: Re-phrase for clarity

Lines 69-70: The characteristics in parentheses in line 70, should be contained within the parentheses in line 69.

Line 109: ‘For the rest of the variables, a one-way ANOVA.’ This is an incomplete sentence and should be merged with the subsequent sentence.

Lines 111-119: This section needs to be re-written for clarity ensuring that the sentence structure and punctuation is correct, currently this is rather poor.

Results: Please also include the magnitude of the differences, rather than simply if they were significant

Line 130: Delete ‘differences’

Lines 134-137: Please re-phrase to clearly state which was greatest and which quarters this was significantly greater than

Lines 147-148: Please state exactly what the differences were, when were they greatest?

Figure 2 requires refinement. The top 2 panels do not require decimal places on the vertical axis. Titles should be removed, and they should be labelled A), B), C) and then defined within the figure legend. Standard deviations should also be presented where appropriate.

Lines 168-172: Please re-write for clarity and be precise about the observed differences, as previously mentioned.

Lines 183-185: Please re-write for clarity and be precise about the observed differences, as previously mentioned.

Lines 190-194: Merge these into one concise sentence

Figure 4: See comments for figure 2

Discussion: All new text needs to be revised for clarity, ensuring the correct use of English throughout.

Line 282: Do not start a new sentence with ‘And…’

Author Response

ALL NEW CORRECTIONS IN THE TEXT APPEAR BLUE FOR BETTER UNDERSTANDING

Open Review 1

English language and style

(x) Extensive editing of English language and style required 
( ) Moderate English changes required 
( ) English language and style are fine/minor spell check required 
( ) I don't feel qualified to judge about the English language and style 

Yes

Can be improved

Must be improved

Not applicable

Does the introduction provide sufficient   background and include all relevant references?

( )

(x)

( )

( )

Is the research design appropriate?

(x)

( )

( )

( )

Are the methods adequately described?

( )

(x)

( )

( )

Are the results clearly presented?

( )

( )

(x)

( )

Are the conclusions supported by the   results?

( )

(x)

( )

( )

Comments and Suggestions for Authors

The authors have made substantial revisions to the manuscript; however, the use of English and the scientific style of writing require attention throughout. Unfortunately, the errors relating to the scientific style of writing and use of English make this difficult to follow and result in some key points being lost.  Please see specific comments below.

We thank the reviewers for their useful suggestions and agree that some of our expression may make our manuscript difficult to follow at some points. In consequence, we have modified  inappropriate expressions and rewrote incorrect paragraphs in order to give a more comprehensive information about our study.

Line 31: The term ‘explosive’ should not be used, please see, Winter et al. (2016). Misuse of "Power" and other mechanical terms in Sport and Exercise Science Research. The Journal of Strength & Conditioning Research 30(1): 292-300.

high insensity

Line 36: Change ‘of’ to ‘on’

Done

Line 60: Delete, ‘For all this, to the best knowledge…’ as it is redundant and makes no sense as part of this sentence.

Done

Line 63-63: Re-phrase for clarity

We have rewritten the sentece: “Besides, accelerations length and intensity will decrease as the game progresses.

Lines 69-70: The characteristics in parentheses in line 70, should be contained within the parentheses in line 69.

Done: (Guards: n= 13, 168.62±5.94 cm; Forwards: n= 22, 176.87±6.04 cm; Centers: n= 13, 183.77±4.71 cm)

Line 109: ‘For the rest of the variables, a one-way ANOVA.’ This is an incomplete sentence and should be merged with the subsequent sentence. Lines 111-119: This section needs to be re-written for clarity ensuring that the sentence structure and punctuation is correct, currently this is rather poor.

We have rewritten all paragraph: “For the rest of the variables, a one way ANOVA was performed. Differences between groups (quarters and playing position) were extensively identified with the Bonferroni post-hoc test. The effect size according to Cohen’s d, was used to identify the differences between groups, considering effect sizes of 0.20 as small, 0.50 as medium and 0.80 as large [13]. Magnitude Based Inferences (MBI) was also calculated to assess the true value of an effect statistic [14]. In cases where the confidence interval overlapped threshold for substantial positive and negative values (±0.20 standardized units), the effect was considered unclear. Otherwise effects were deemed clear. Thereby, results were labelled as probabilities, being probable, very probable and almost certain values identified as the most important effects in practice

Results: Please also include the magnitude of the differences, rather than simply if they were significant

Done

Line 130: Delete ‘differences’

Done

Lines 134-137: Please re-phrase to clearly state which was greatest and which quarters this was significantly greater tan

Done: There were statistically significant differences in the intensity of the accelerations and decelerations between quarters (ACC: χ² =39.608; V Cramer=.051; p=.000; DEC: χ² =24.696; V Cramer=.073; p=.000), being higher in the last quarter compared to the rest (Figure 1).

Lines 147-148: Please state exactly what the differences were, when were they greatest?

Done, we hace rewritten al paragraph: “The accelerations were longer in the first period (Q1= 2138.31 ms; Q2= 2157 ms) than the second period (Q3= 2035.27 ms; Q4= 2004.58 ms) with Almost certain magnitude-based inferences (p=.001; MBI= 99.8). The decelerations were longer in the first period (Q1= 2005.64 ms; Q2= 2033.90 ms) than the second period too (Q3= 1897.33 ms; Q4= 1912.57 ms) with Almost certain magnitude-based inferences (p=.001; MBI= 99.3-99.8). The accelerations start speed was greater in the first period (Q1= 2.54 km/; Q2= 2.51 km/) than the second period (Q3= 2.34 km/h; Q4= 2.41 km/h) but significant differences were found in the comparison with the third quarter (Q1: p=.003 and MBI=98.1; Q2: p=.001 and MBI=99.3). ). The decelerations start speed was greater in the first period (Q1= 12.13 km/h; Q2= 11.93 km/h) than the second period too (Q3= 11.41 km/h; Q4= 11.47 km/h) with Almost certain magnitude-based inferences (p=.001; MBI= 99.7-99.9). The smallest acceleration peak occurred in Q3 (2.34 m/s2) compared to Q1 and Q2 (p=.003 and MBI=99.3; p=.002 and MBI=99.5).

Figure 2 requires refinement. The top 2 panels do not require decimal places on the vertical axis. Titles should be removed, and they should be labelled A), B), C) and then defined within the figure legend. Standard deviations should also be presented where appropriate.

Done

Lines 168-172: Please re-write for clarity and be precise about the observed differences, as previously mentioned.

We have rewritten all paragraph: “The statistically significant differences were effectively evaluated by the magnitude analysis. “Almost certain” values were identified as the most important effects. These effects were mainly observed in the length of accelerations and decelerations, being substantially higher in the first period (Q1 and Q2). Specifically, there were important effects in decelerations in the start speed, being also higher in the first period.

Lines 183-185: Please re-write for clarity and be precise about the observed differences, as previously mentioned.

Done: “Therefore, there were statistically significant differences in the accelerations and decelerations intensity ranges between playing position (ACC: χ² =81.230; V Cramer=.073; p=.000; DEC: χ² =95.632; V Cramer=.081; p=.000) being higher in Guards compared to the rest”

Lines 190-194: Merge these into one concise sentence

We have rewritten all paragraph: “The results show statistically significant differences in the lenght of accelerations, maximum speed peak and start speed (p<.005) among playing positions. There were differences in the length of accelerations between center and perimeter players (ACC=2200.87 ms; DEC=2050.87 ms) but no important effect was observed. The maximum speed peak and start speed were higher between Guards (ACC=4.55 m/s2 and 2.70 km/h) and the rest playing positions with almost certain magnitude-based inferences (p= .001 and MBI= 99.8 – 99.9). In the variables of deceleration, no important effects were observed (Figure 4).”

Figure 4: See comments for figure 2

Done

Discussion: All new text needs to be revised for clarity, ensuring the correct use of English throughout.

We have modified  inappropriate expressions and rewrote incorrect paragraphs in order to give a more comprehensive discussion.

Line 282: Do not start a new sentence with ‘And…’

Done

Reviewer 2 Report

Minor comments

The authors have replied in a satisfactory way to most of the comments made.

The following minor changes should be done:

Line 16: change “collaborative-opposition sports” to “team sports”.

Line 33: Please clarify what you mean by “collective sports”

Line 60: delete the words: “For all this, to the best knowledge”

Line 70: replace comma with “.” for decimal points

Fig. 2 (top figure): use no decimal points for ms

Language must be thoroughly checked and corrected throughout the manuscript, as there are many expression errors.

Author Response

ALL NEW CORRECTIONS IN THE TEXT APPEAR BLUE FOR BETTER UNDERSTANDING

Open Review 2

English language and style

( ) Extensive editing of English language and style required 
(x) Moderate English changes required 
( ) English language and style are fine/minor spell check required 
( ) I don't feel qualified to judge about the English language and style 

Yes

Can be improved

Must be improved

Not applicable

Does the introduction provide sufficient   background and include all relevant references?

( )

(x)

( )

( )

Is the research design appropriate?

( )

(x)

( )

( )

Are the methods adequately described?

( )

(x)

( )

( )

Are the results clearly presented?

( )

(x)

( )

( )

Are the conclusions supported by the   results?

( )

(x)

( )

( )

Comments and Suggestions for Authors

Minor comments

The authors have replied in a satisfactory way to most of the comments made.

The following minor changes should be done:

Line 16: change “collaborative-opposition sports” to “team sports”.

Done

Line 33: Please clarify what you mean by “collective sports”

Done

Line 60: delete the words: “For all this, to the best knowledge”

Done

Line 70: replace comma with “.” for decimal points

Done

Fig. 2 (top figure): use no decimal points for ms

 Done

Language must be thoroughly checked and corrected throughout the manuscript, as there are many expression errors.

We thank the reviewers for their useful suggestions and agree that some of our expression may make our manuscript difficult to follow at some points. In consequence, we have modified  inappropriate expressions and rewrote incorrect paragraphs in order to give a more comprehensive information about our study.

Round  3

Reviewer 1 Report

General Comments

The authors have made substantial improvements to the manuscript, however, careful proof reading is still required to ensure that the level of English is sufficient for publication, including sentence structure and revision  of numerous fragmented sentences. Some of these errors are highlighted in the specific comments below.

Specific Comments

Line 37: ‘…final performance…’ does not appear to make sense here

Line 63: This is fragmented and needs to be refined or merged with the previous sentence.

Line 109: Delete ‘extensively’

Line s110-111: Please re-write for clarity and present the categories appropriately, as ranges rather than exact values, for example, if >0.80 is large, then 0.50-0.80 would be medium and 0.20-0.49 would be small, but what if the values are<0.20, should this be trivial?

Line 112: Change ‘was’ to ‘were’

Line 165: This sentence does not make sense and is fragmented. In addition, MBI’s are inanimate and therefore cannot evaluate anything; please consider that ‘MCI’s were used to evaluate…’

Line 166: Merge the sentences

Line 167-168: Please re-write this sentence for clarity

Line 184-185: As mentioned in the previous revision, do not state differences, this is too vague, which were greater / greatest?

Line 220: Change ‘was’ to ‘were’

Line 221: Delete ‘According to plying position…’ as this is redundant here

Line 231: Please re-write this sentence for clarity

Line 232: This sentence appears to contradict the previous sentence, please ensure clarity across these sentences.

Line 235: Please merge these sentences

Line 237: Please merge these sentences

Author Response

Open Review

English language and style

(x) Extensive editing of English language and style required 
( ) Moderate English changes required 
( ) English language and style are fine/minor spell check required 
( ) I don't feel qualified to judge about the English language and style 

Yes

Can be improved

Must be improved

Not applicable

Does the introduction provide sufficient background   and include all relevant references?

( )

(x)

( )

( )

Is the research design appropriate?

(x)

( )

( )

( )

Are the methods adequately described?

(x)

( )

( )

( )

Are the results clearly presented?

( )

(x)

( )

( )

Are the conclusions supported by the results?

( )

(x)

( )

( )

Comments and Suggestions for Authors

General Comments

The authors have made substantial improvements to the manuscript, however, careful proof reading is still required to ensure that the level of English is sufficient for publication, including sentence structure and revision  of numerous fragmented sentences. Some of these errors are highlighted in the specific comments below.

WE HAVE CONTINUED TO MODIFY INAPPROPRIATE EXPRESSIONS AND HAVE REWRITTEN INCORRECT WORDS AND PARAGRAPHS TO PROVIDE MORE COMPLETE INFORMATION ABOUT OUR STUDY. IN ADDITION, WE ATTACH A CERTIFICATE OF REVIEW BY AN EXPERT IN THE FIELD

Specific Comments

Line 37: ‘…final performance…’ does not appear to make sense here

WE HAVE MODIFIED THE SENTENCE: These actions are crucial in the athlete’s performance in team sports”

Line 63: This is fragmented and needs to be refined or merged with the previous sentence.

WE HAVE SIMPLIFIED THE SENTENCE: “Therefore, it was hypothesized that accelerations length and intensity will decrease as the game progresses.

Line 109: Delete ‘extensively’

DONE

Line s110-111: Please re-write for clarity and present the categories appropriately, as ranges rather than exact values, for example, if >0.80 is large, then 0.50-0.80 would be medium and 0.20-0.49 would be small, but what if the values are<0.20, should this be trivial?

DONE

Line 112: Change ‘was’ to ‘were’

DONE

Line 165: This sentence does not make sense and is fragmented. In addition, MBI’s are inanimate and therefore cannot evaluate anything; please consider that ‘MCI’s were used to evaluate…’

Line 166: Merge the sentences

Line 167-168: Please re-write this sentence for clarity

WE HAVE MODIFIED ALL THE PARAGRAPH TO ANSWER YOUR LAST THREE RECOMMENDATIONS: “MBI’s were used to evaluate the effect statistic [14], the results labeled as “Almost certain” values were identified as the most important effects. “Almost certain” effects were observed in the length of accelerations and decelerations, being substantially higher in the first period (Q1 and Q2) compared to the second period (Q3 and Q4). In the start speed, there were important effects in decelerations, being higher in the first period compared to the rest of periods.

Line 184-185: As mentioned in the previous revision, do not state differences, this is too vague, which were greater / greatest?

Dear author, thank you for your insistence but we understand that this suggestion has already been resolved. The first sentence has the aim to introduce the following results "The results show statistically significant differences in the length of accelerations, maximum speed peak and start speed (p <.005) among playing positions". Next, we detail exactly where we find those differences. For example, “There were differences in the length of accelerations between center and perimeter players being greater in Guards and Forwards”; “The maximum speed peak and start speed were higher in Guards than the rest playing positions”

Line 220: Change ‘was’ to ‘were’

DONE

Line 221: Delete ‘According to plying position…’ as this is redundant here

DONE

Line 231: Please re-write this sentence for clarity

WE HAVE MODIFIED THIS SENTENCE: “In the analysis of game periods, no significant differences were found in the number of accelerations and decelerations performed by the players but their intensity changed, being higher in the last quarter.”

Line 232: This sentence appears to contradict the previous sentence, please ensure clarity across these sentences.

We don’t think it is contradictory: the NUMBER of accelerations and decelerations is similar in the four periods, however the INTENSITY of these accelerations is greater in the last

Line 235: Please merge these sentences

DONE

Line 237: Please merge these sentences

DONE